Subject Areas:
computational biology/computer modelling and simulation/theoretical biology

Keywords:
genetic drift, natural selection, population structure, fixation, network, recombination

Author for correspondence:
Peter A. Whigham
e-mail: peter.whigham@otago.ac.nz

# Graph-structured populations and the Hill–Robertson effect

Peter A. Whigham[1] and Hamish G. Spencer[2]

[1]Department of Information Science, and [2]Department of Zoology, University of Otago, PO Box 56, Dunedin 9054, New Zealand

 PAW, 0000-0002-8221-6248; HGS, 0000-0001-7531-597X

The Hill–Robertson effect describes how, in a finite panmictic diploid population, selection at one diallelic locus reduces the fixation probability of a selectively favoured allele at a second, linked diallelic locus. Here we investigate the influence of population structure on the Hill–Robertson effect in a population of size $N$. We model population structure as a network by assuming that individuals occupy nodes on a graph connected by edges that link members who can reproduce with each other. Three regular networks (fully connected, ring and torus), two forms of scale-free network and a star are examined. We find that (i) the effect of population structure on the probability of fixation of the favourable allele is invariant for regular structures, but on some scale-free networks and a star, this probability is greatly reduced; (ii) compared to a panmictic population, the mean time to fixation of the favoured allele is much greater on a ring, torus and linear scale-free network, but much less on power-2 scale-free and star networks; (iii) the likelihood with which each of the four possible haplotypes eventually fix is similar across regular networks, but scale-free populations and the star are consistently less likely and much faster to fix the optimal haplotype; (iv) increasing recombination increases the likelihood of fixing the favoured haplotype across all structures, whereas the time to fixation of that haplotype usually increased, and (v) star-like structures were overwhelmingly likely to fix the least fit haplotype and did so significantly more rapidly than other populations. Last, we find that small ($N < 64$) panmictic populations do not exhibit the scaling property expected from Hill & Robertson (1966 *Genet. Res.* **8**, 269–294. (doi:10.1017doi:10.1017/S0016672300010156)).

## 1. Introduction

Spatial structure is a defining characteristic of all biological systems. Gene flow within a finite structured population affects the trajectory of an evolving system, most notably the probability and time to fixation of an allele. These two properties of fixation are fundamentally linked to the actions of

genetic drift, selection, recombination, as well as the differential availability of mates arising from the structure of the population. This last aspect—the way in which the population is subdivided – has long been of interest to population-genetic theorists [1–7]. Of note is early work by Maruyama [3,8] and Nagylaki [9] that showed under certain conditions fixation probability at a single locus subject to selection was invariant to the form of subdivision of an island-based population with migration. This invariance property will be addressed in terms of regular graphs in §3.2.

Most of these studies have modelled population structure as a number of subpopulations (conceptually, islands) connected in various ways by the exchange of migrant individuals. In the last 15 years, however, various authors [10–17] have modelled population structure as a graph, by assuming that individuals in the population sit at vertices on networks that are connected (via edges) to other vertices in various ways. Interactions between individuals (such as mating) occur only if their vertices are connected by an edge. Such an approach can be seen as a significant generalization of the traditional island models, since the latter can always be represented as graphs.

Although much population-genetic theory concerns just a single locus, the properties of two-locus models are also of great interest, not the least because the behaviour of such models often cannot be inferred from the simpler one-locus models. For example, Hill & Robertson [18] demonstrated that in a finite, panmictic diploid population, selection at one locus reduces the fixation probability of a selectively favoured allele at a second, linked locus, a behaviour that became known as the Hill–Robertson (HR) effect [19]. This property of linked loci had significant theoretical implications including providing arguments for the evolutionary role of recombination [19], the role of linkage in codon bias [20], Muller's ratchet [21] and structural features observed in real genomic data [22]. The model developed by Hill & Robertson [18] was motivated by initial work of Kimura [23], who showed that the probability of fixation for an allele at a single locus in a randomly mating population is dependent only on the initial frequency of the allele, $p_0$, and a compound parameter $Ns$, where $N$ is the population size and $s$ is the difference in selective advantage between the two homozygotes (with heterozygotes having an intermediate advantage of $1/2s$). Extending this concept, Hill and Robertson constructed a model similarly scaled by $N$, in which the selective advantages at the two loci were $\alpha$ and $\beta$, respectively, and $c$ was the probability of crossing over between the loci. The simulations of Hill and Robertson also appeared to show that if selection and recombination are scaled by population size, $N$, there is no effect of $N$ on the probability of fixation. Finally, they showed that the time to fixation (expressed as the half-life) decreases with tighter linkage. We refer the reader to the review by Comeron *et al.* [24] for further details.

Here we use a network approach to extend the classical work of Hill & Robertson [18] to address the issue of how spatial structure alters fixation probability and time. Specifically, we used computer simulations over a range of population structures to examine the probability of and time to fixation of a selectively favoured allele at a locus linked to another at which selection is also acting. We examine how these two properties are affected by population size ($N$), selection strength ($N\alpha$, $N\beta$) and crossover (recombination) rate ($Nc$). Given that population structure modelled as a network affects a single-locus diploid model of genetic drift [11], it is not surprising that we find similar consequences for two linked diploid loci subject to selection and drift.

Although the use of network structures for examining how evolutionary dynamics vary with population structure is now common in theoretical biology, the main emphasis has been with the birth/death processes modelled as a Moran process, game theory models and the evolution of cooperation. For example, Nowak *et al.* [25] applied game theory to model competition between species, while Allen *et al.* [26] provided a mathematical treatment using coalescence times for graph structures under weak selection. Other work has examined the type of graph structures that lead to amplification of selection [27], the properties of fixation probability and time for the Moran process [10,13,14,16,28], mathematical models for fixation time for specific k-partite graphs [29] and approaches to approximate the fixation probability for neutral drift models [30]. This field of research has emphasized game-theoretic models for behaviour, or used a two-allele haploid model based on the Moran process, with the emphasis on how the evolution of cooperation or fixation of a mutant is altered by population structure. Other work has examined the influence of population structure on Muller's ratchet [31]. It is worth noting that these models all use a single population with overlapping generations, normally replacing just one (or occasionally, two) individuals per time step (described as a steady-state model [32]). The generational model, mainly used in evolutionary biology and specifically in the HR model [16], replaces an entire population each generation. In the electronic supplementary material, we show that this distinction is crucial in terms of how spatial structure affects both fixation probability and time. Although this effect has previously been described in detail

[33], the concepts are revisited in the electronic supplementary material, section S2, to verify the results for more complex genetics model.

To our knowledge, no previous work has examined the influence of an evolving diploid population with linkage disequilibrium for a broad range of population structures represented as a graph. Of note, however, are the results of [10] where it was shown that fixation probability for a single introduced mutant under selection for the Moran process is the same for all isothermal graphs (in which each vertex has the same degree and sum of weighted links represented as a doubly stochastic matrix), even though fixation times may vary. A similar empirical result under a generational process is presented here.

The influence of spatial segregation on the HR effect, based on an island model with migration, has been previously considered. Reeve *et al*. [34] examined the evolution of recombination rates in two finite populations (islands) with migration where each population was adapting to different environments. The role of linkage disequilibrium and the HR effect was shown to favour increased recombination early in the evolution, but with subsequent migration by maladapted migrants between the two population, the cost of recombining with these ill-suited individuals meant reduced recombination rates were favoured later in the evolutionary process.

Nevertheless, the majority of work that examines the HR effect has not considered spatial structures, instead focusing on the consequences of weak selection and linkage in finite populations and the implications for panmictic populations.

# 2. Methods

The model representation and terminology largely follows Hill & Robertson [18], which allows a direction comparison between their results and ours. We model individuals as monoecious, diploid individuals, and follow the dynamics of two alleles at each of two selectively additive linked loci in a finite population. There are thus four possible gametes: *AB*, *Ab*, *aB* and *ab*. The initial frequencies of the *A* (*a*) and *B* (*b*) alleles are denoted by $p_0$ ($p_1 = 1 - p_0$) and $q_0$ ($q_1 = 1 - q_0$), respectively. Given a $p_0$ and $q_0$, the population was initialized randomly (i.e. assuming no linkage disequilibrium, as in [18]). Although some previous work [35] has examined the effect of initial linkage disequilibrium on fixation probability, for our comparative work this was not considered. The rate of recombination is *c* and the selective strength for the *A* allele is *α* and that for *B* allele is *β*, with selection acting independently at each locus. So, for example, the fitness of an *Ab/ab* individual is $(1 - 1/2\alpha)(1 - \beta)$.

In contrast to Hill & Robertson [18], who model an unstructured or panmictic population, we investigate a range of population structures. The structure for a population of *N* individuals was represented as a connected network (graph) **G** with *N* locations (nodes), following previous work in modelling generalized spatial structure [10,11]. This structure is represented as a symmetric $N \times N$ weighted connectivity matrix **M**, where the entries $m_{ij} = m_{ji} > 0$, indicate that individuals $N_i$ and $N_j$ are part of the same subpopulation (deme) and $m_{ij} = m_{ji} = 0$ indicates that they are in separate demes. Each individual (location) was connected to itself ($m_{ii} > 0$) since a node represents a location to construct a deme and an individual at a node (location) is always part of that deme. The entries of **M** represent the weightings of the edges connecting each location. Following [10], **M** was designed as a stochastic matrix, meaning that the rows of **M** sum to one. Spatial structure did not change across generations and we followed a generational mating process. Given a population of *N* individuals occupying a graph $\mathbf{G_g}$ at generation *g* with connectivity matrix **M,** the next generation, $g + 1$, was created as follows:

For each location (node) $N_i \in G_g$

1. Create a subpopulation of individuals $N_{is} = \{N_j \mid m_{ij} > 0\}$.
2. Select (with replacement) two parents $P_1$ and $P_2$ from $N_{is}$ each with probability proportional to the product of the weight of the edge connecting them to $N_i$ and their relative fitness in the subpopulation $N_{is}$, as suggested by Lieberman *et al*. [10].
3. Generate gametes, one each from $P_1$ and $P_2$, with recombination rate *c*.
4. Create child $C_i$ by uniting these gametes.
5. Insert $C_i$ at location $N_i$ for generation $G_{g+1}$.

Once all locations have been visited, $G_{g+1}$ replaces the current generation $G_g$. Each replication in our simulations was run until both loci fixed. The number of generations until fixation of *A/a* and *B/b*

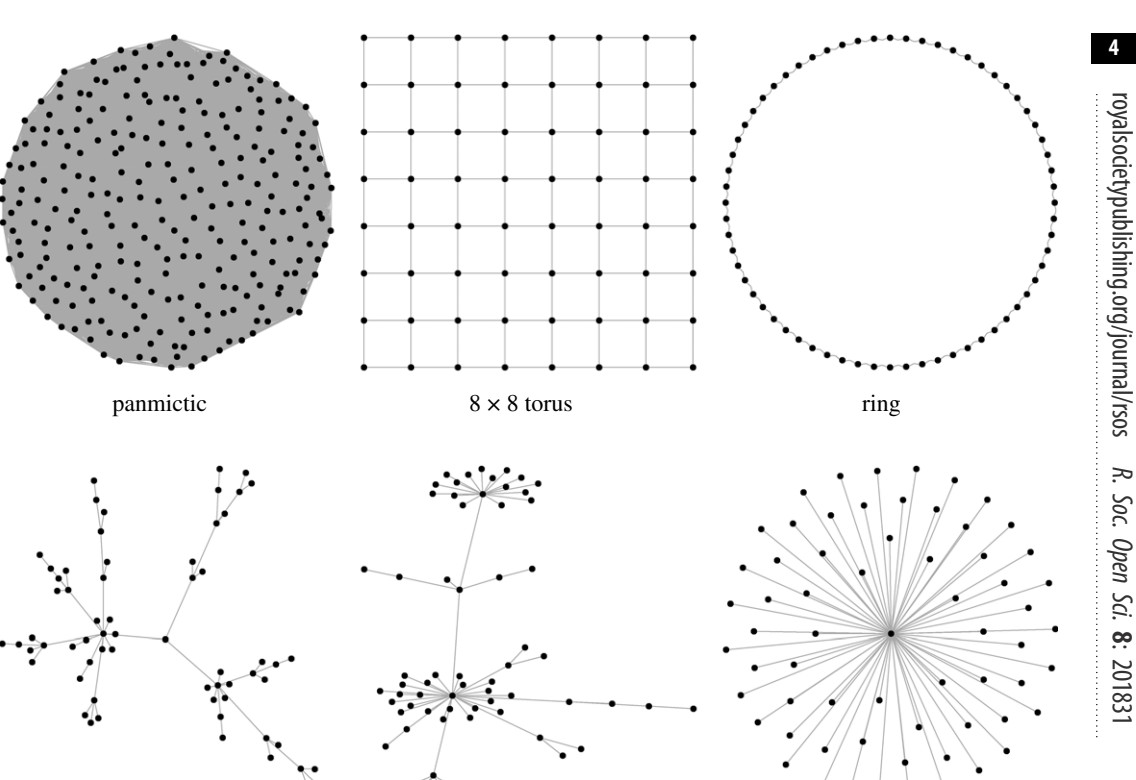

**Figure 1.** Example networks ($N = 64$). The panmictic, torus and ring networks have regular structure with all nodes having the same degree. The scale-free networks show hub-like features, while the star has a single, centralized hub.

were recorded. All runs (except where stated) were performed for $N = 64$. For each set of parameters (namely, $p_0$, $q_0$, $\alpha$, $\beta$ and $c$), 10 000 independent replicates were run; these runs differed only in the pseudo-random numbers used to select parents and determine gamete composition.

The following graph structures were used for all simulations: a fully connected, equally weighted (panmictic) graph; a ring structure where each location is connected to their four nearest neighbours (i.e. two on each side); an $8 \times 8$ torus where each location connects to four neighbours and edges wrap around, two forms of scale-free network [36] with the power of preferential attachment set to one or two; and a star network with a single, centralized node. The regular graphs (fully connected, ring and torus) were constrained to have $\mathbf{M}$ as a doubly stochastic matrix, while the scale-free and star networks set outward edge weights to 1/degree(node), meaning $\mathbf{M}$ was a right stochastic matrix. The scale-free graphs were sampled anew for each replicate simulation using the R package *igraph* [37], since the graph generators were stochastic. Example graphs are shown in figure 1.

To biologists the graph structures used in this paper may seem arbitrary and unrealistic. However, the structures are chosen to highlight and exemplify different population constraints. For example, the ring and star are extreme examples of a regular structure and single clustered population, while the scale-free graphs represent linear structures with increased local clustering. The purpose here is to emphasize how changing population structures affect and interact with linked loci. We acknowledge that biological populations will only present elements of these differences due to the complexity of a real population.

# 3. Results

## 3.1. Model validation: panmictic populations

We confirmed a number of Hill and Robertson's [18] results with our formulation of the model as part of our model verification. For example, figure 2 shows, for a range of panmictic populations of size $N$, the frequency of fixation of the $A$ allele at the first locus for a range of initial frequencies of $B$ at the second locus ($q_0$), for a fixed set of selection and crossing parameters, and should be compared to the lines for

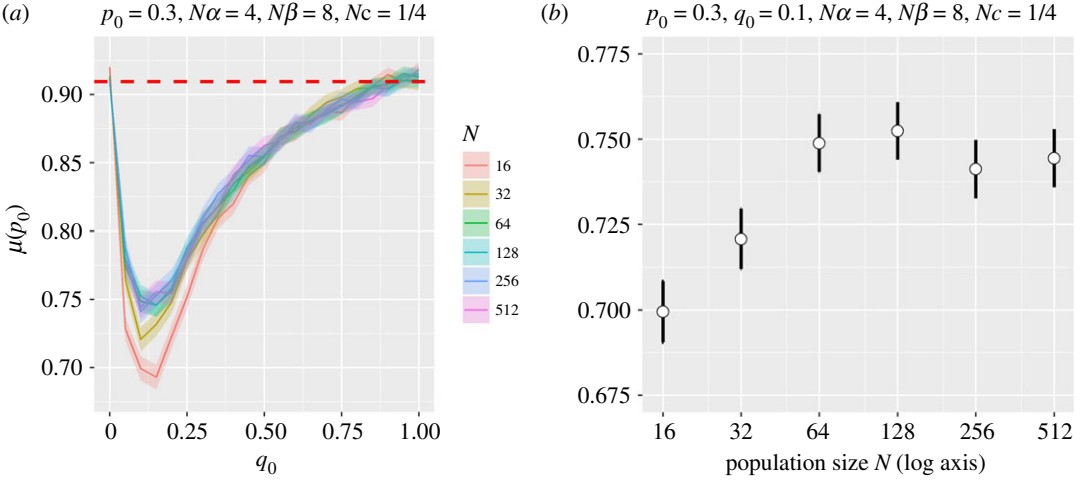

**Figure 2.** Panel (a) shows fixation probability at the first locus for allele A for a range of different panmictic population sizes (N) and the initial frequency of allele B at the second locus. Otherwise, $p_0 = 0.3$, $N\alpha = 4$, $N\beta = 8$ and $Nc = 1/4$. The dashed line shows the Kimura approximation for a single locus with the same $p_0$ and $N\alpha$ values. Panel (b) shows variation for $p_0 = 0.3$ and $q_0 = 0.1$ with 95% confidence intervals.

$Nc = 1/4$ in Hill and Robertson's fig. 1. The 95% confidence intervals about the mean were calculated by creating a variable F, where F = {1 | A fixed, 0 otherwise} for each run and using a t-distribution over F

$$\text{confidence interval} = \mu(F) \pm qt(0.975, |F| - 1) * \frac{sd(F)}{\sqrt{|F|}} ,$$

where $|F|$ is the number of values of F, $qt(0.975, df)$ is the t-distribution quantile function for a 2.5% interval given df degrees of freedom, and $sd(F)$ is the standard deviation for F.

In contrast to Hill and Robertson's claim [18], there is some evidence that the fixation probability is not independent of N when scaled: the values are lower for the lowest value of N (N = 16). Nevertheless, all runs show the same overall pattern of a rapid drop in the probability of fixation as $q_0$ increases from 0.0 to 0.1, followed by an almost equally rapid increase as $q_0$ increases further. We note that when $q_0 = 0$ or 1, the probabilities are approximately identical for each N, a consequence of the lack of variation at locus B and the resultant reduction of the problem to one at a single locus. Indeed, Kimura's approximation [23], which should be more accurate for larger population sizes, gives this probability (see dashed line, figure 2) as

$$\mu(p_0) = \frac{1 - e^{-2N\alpha p_0}}{1 - e^{-2N\alpha}} = \frac{1 - e^{-2.4}}{1 - e^{-8}} \approx 0.9096.$$

Apart from N = 16 and 32, the behaviours are consistent across N. Since we require N to be large enough to populate a variety of population structures, these initial validation experiments suggested that N = 64 is a suitable choice for our remaining simulations.

The sensitivity to N (i.e. the contradiction of Hill and Robertson's claim) is even clearer in electronic supplementary material, figure S1, which has stronger selection at locus A and initially a rarer allele A (i.e. a smaller $p_0$). In this case, in contrast to that in figure 2, smaller populations appear more likely to fix allele A, although the effect diminishes as N increases. These differences are a result of the changing balance between drift and selection for small populations. With these parameters, Kimura's one-locus formula gives the probability as

$$\mu(p_0) = \frac{1 - e^{-1.6}}{1 - e^{-16}} \approx 0.7981,$$

which corresponds well to the simulation values for $q_0 = 0$ or 1 when $N \geq 64$.

A further justification for our choice of N is adumbrated in the supplementary material with electronic supplementary material, figure S2. Here the fixation probability for a regular square torus is shown for sizes $8 \times 8$, $12 \times 12$, $16 \times 16$ and $20 \times 20$ for a range of parameter settings. The similarity of fixation probabilities for each N shows that the scaling property hypothesized by HR applies to regular graphs once N is sufficiently large. However, this invariance does not apply to irregular

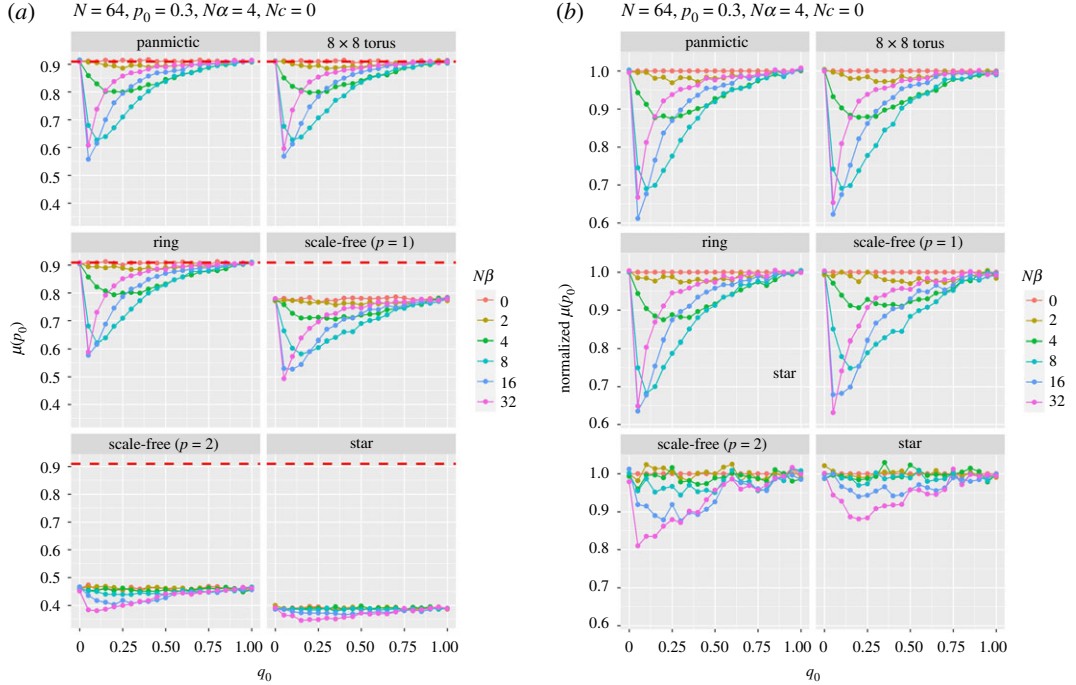

**Figure 3.** Panel (a) shows the probability of fixation of allele A at the first locus as a function of the initial frequency of favoured allele B at the second locus ($q_0$) with different strengths of selection ($\beta$) in favour of allele B. No crossing over. The red dashed line shows the Kimura approximation for a single locus with $p_0 = 0.3$ and $N\alpha = 4$. Panel (b) shows fixation probability normalized by that for $N\beta = 0$ for each $q_0$.

population structures such as scale-free and star networks. This lack of invariance is clearly demonstrated in electronic supplementary material, figures S3 and S4 where scale-free networks ($p = 2$) and star networks of sizes 64, 128, 256 and 512 are shown for a range of parameter settings. Here, as $N$ becomes larger, the hub-like graph structures reduce the influence of selection on fixation probability resulting in a population behaving as if it were subject to drift alone. Since we want to understand how the balance of selection and drift for the HR model are affected by different graph structures we used $N = 64$ so that the effect of selection remains prominent for irregular graphs.

## 3.2. Spatial structure and fixation

Figure 3a shows the probability of fixation of the A allele in the absence of recombination for a range of selection strengths at the second locus ($N\beta$). The panmictic results replicate those in Fig. 1 of Hill & Robertson [18]. The graphs reveal a remarkable insensitivity of this probability to major changes in population structure for regular networks. The torus and ring have almost identical fixation probabilities to panmictic populations, even though the mean path length (ignoring edge weights) on a ring (8.38) and torus (4.1) is significantly greater than a fully connected network (1). Although the linear power scale-free ($p = 1$) has a mean path length of 4.8, there is still a reduced probability of fixing A. Indeed, for some parameter combinations (strong selection at B and allele A being rare), this probability drops below 0.5. The scale-free ($p = 2$, path length 2.52) and star populations (path length 1.97) are significantly more likely to fix the disfavoured allele (a) for all parameter combinations. Interestingly, all simulations show the same shaped curves, although those for scale-free and star populations are shifted down and compressed compared to those for the regular networks. The variance in degree for a graph can be used as a general measure of graph irregularity. For all regular graphs this value is zero, while for scale-free ($p = 1$) degree variance is approximately 3.7, scale-free ($p = 2$) degree variance is approximately 15.6 and for a star degree variance is 60.06. This clearly correlates to the observed changes in fixation probability and implies that graphs with nodes that are much more connected than average are strong influences of fixation.

We also show, in figure 3b, the results scaled by the response when there is no selection at the second locus. Here each graph is normalized by the probability of fixation when $N\beta = 0$ for each $q_0$. Panel b shows that relative HR probabilities are more or less the same for graphs that are regular or have some clustering

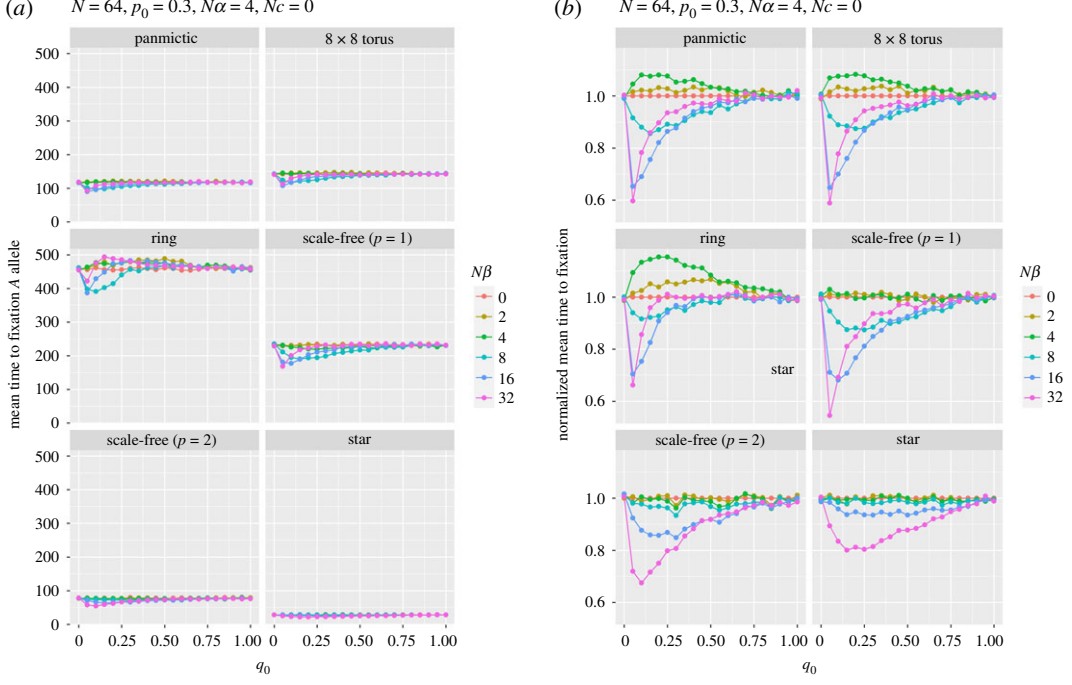

**Figure 4.** Panel (*a*) shows the mean time to fixation at the first locus (A) as a function of the initial frequency of allele *B* ($q_0$) for various strengths of selection (*β*) at the second locus with no recombination. Panel (*b*) shows time to fixation normalized by that for $N\beta = 0$ for each $q_0$.

with linear structures (scale-free $p = 1$). When a small number of nodes have high degrees compared with most other nodes (scale-free $p = 2$ and star); however, the effect of linked selection is reduced, which cannot be accounted for by some sort of effective population-size adjustment. In other words, the effects we see are not simply the result of some change in effective population size: structure matters, too.

In stark contrast, the mean time to fixation is dramatically affected by population structure (figure 4*a*). As Whigham *et al.* [11] found in their models of drift at a single locus in the absence of selection, compared with panmictic populations, mean fixation times are longer on rings and shorter in populations with centralized nodes. This result was first examined in detail by Slatkin [38], where he showed that fixation time increased for both stepping stone and island-based models compared to the equivalent panmictic population. Scale-free worlds, with their central hubs are more likely to fix an allele by drift, even in the presence of selection, and are consequently more likely to fix a disfavoured allele. Moreover, if the disfavoured allele (*a*) is going to fix, it must do so quickly, before selection has time to increase the initial frequency of *A* ($p_0 = 0.3$). The scale-free ($p = 1$) population has a slower fixation time than a panmictic population, but a reduced fixation of allele *A* (figure 3). These two properties can be explained in terms of the linear structures that connect centralized hubs for the scale-free ($p = 1$) networks (figure 1) resulting in longer fixation times, but allowing the centralized hubs to fix the disfavoured allele. The single centralized hub of the star reduces the fixation time to approximately 25% of the panmictic population.

Figure 4*b* shows normalized fixation time, which shows two properties: as the path length of the graph increases (e.g. as it does in the ring), the relative fixation time increases for balanced selection ($N\alpha = N\beta$); for the scale-free ($p = 2$) and star graphs the effect on fixation time due to different selection strengths is reduced. Again, these results show that a simple adjustment to effective population size would not account for the differences in fixation time. Again, structure matters, influenced by both graph path length and degree variance.

In all these simulations, regardless of strength of selection and the structure of the population, the shortest mean time to fixation occurred when the initial frequency of the allele *B*, $q_0$, was small (most often 0.05, 0.10 or 0.15), which was also when the probability of fixing the favoured allele *at the other locus* (A) was smallest. This effect was strongest when selection at locus B was stronger. A low (but non-zero) frequency of *B* means that haplotype *Ab* is commoner than the relatively fitter *AB*. Hence, the frequency of allele *A* is reduced by selection against *b*, in a classical Hill–Robertson effect that sees

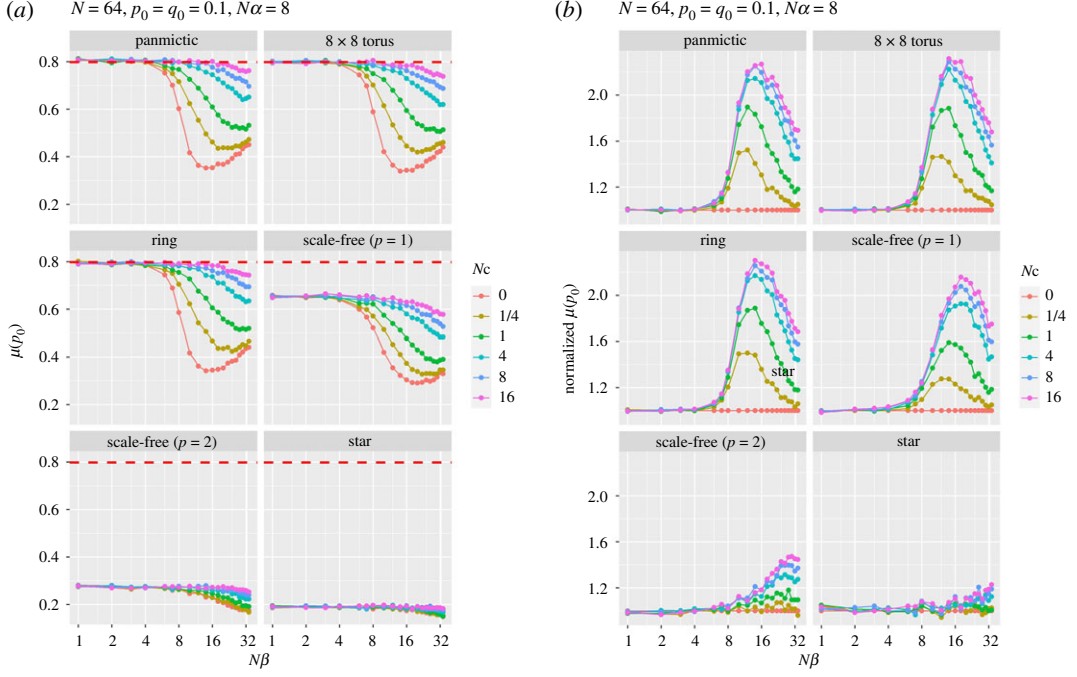

**Figure 5.** Panel (*a*) shows the probability of fixation at the first locus (A) as a function of various selection strengths at the second locus (B) for a range of recombination rates, *Nc*. Panel (*b*) shows fixation probability normalized by that for *Nc* = 0 for each value of *Nβ*.

the frequency of *aB* increase (made clearer by the absence of recombination in these simulations). And, obviously, this effect is greater with stronger selection at locus B.

Mean fixation time is longer than panmictic populations for the torus, ring and scale-free networks with linear attachment (*p* = 1), and shorter for a scale-free (*p* = 2) and star networks. Mean time to fixation is reduced for larger *Nβ* and small initial frequency *q₀*. Similar explanations pertain to figures 5 and 6, which show, respectively, the probability of fixation (and normalized probability) of allele *A* and the mean time to that fixation (and normalized fixation time), for a range of population structures and recombination rates, as a function of the strength of selection at locus B. The curves for the probabilities of fixation for panmictic, torus and ring are strikingly similar, although the times are not, with ring, torus and scale-free (*p* = 1) graphs being slower, reflecting the average path lengths. The scale-free (*p* = 2) and star simulations show, as before, the strong effect of their central hubs. In none of these runs was the favoured allele (*A*) more likely to fix than allele *a*.

Interestingly, as can be seen in figure 6, for all spaces except scale-free (*p* = 2) and star, when there is no recombination the time to fixation increases until selection is approximately of equal strength at the two loci. Further increases in the strength of selection at locus B give much shorter fixation times, which then lengthen as selection becomes even stronger. When selection at the two loci is approximately equal, the lack of recombination means that the fittest haplotype (*AB*), which is initially very rare, remains so for several generations (and, indeed, may even become extinct through genetic drift) because the prime beneficiaries of selection are the *Ab* and *aB* haplotypes. As selection at the B locus becomes stronger, fixation at A happens faster, but is more likely to fix allele *a*. This pattern is roughly repeated, albeit less strikingly, for small recombination rates.

Figures 5*b* and 6*b* show the normalized response to fixation and time to fixation by dividing the responses by the values for *Nc* = 0 for each *Nβ*. Both show that the extreme degree structures of the scale-free (*p* = 2) and star result in responses that cannot be accounted for by effective population-size adjustments alone. Interestingly, for these graphs when *Nβ* ≤ *Nα* normalized fixation time is unaffected, irrespective of recombination rate.

When the strength of selection at locus A was halved (electronic supplementary material, figures S5 and S6), the mean time to fixation was remarkably robust to changes in the strength of selection at locus B and the level of recombination. Nevertheless, the ranking of different structures, with ring, torus and scale-free (*p* = 1) being slowest and the scale-free (*p* = 2) and star models fastest, remained the same. As would be expected, the probability of fixing allele *A* was most sensitive to changes in the strength

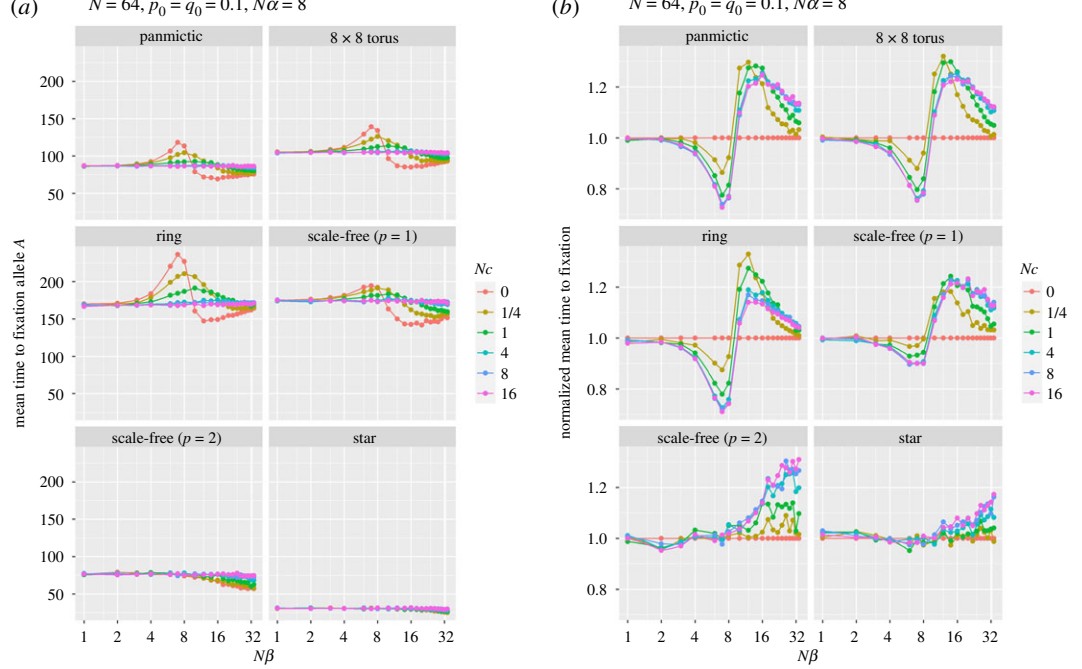

**Figure 6.** Panel (*a*) shows the mean time to fixation at the first locus (A) as a function of various selection strengths at the second locus (B) for a range of recombination rates *Nc*. Panel (*b*) shows fixation time normalized by that for *Nc* = 0 for each *Nβ*.

of selection at locus B when recombination was zero. Stronger selection at locus B at first reduced the chance of fixing allele *A* before even stronger selection increased this chance, although never to the point when it was more likely than in the absence of selection at locus B.

## 3.3. Haplotype fixation and population structure

Figure 7 shows the fixation probability of the four possible haplotypes for different strengths of selection at locus B and recombination rates *Nc*. The results for ring and torus networks were almost identical to those for panmictic populations, and are not shown. In numerous cases for the panmictic and scale-free (*p* = 1) populations, haplotype *Ab* was more likely to fix than the best haplotype (*AB*), but generally stronger selection in favour of *B* and looser linkage were, not surprisingly, more likely to result in the fixation of *AB*. For weak selection at *B* (*Nβ* = 2), the least fit—but initially most frequent—haplotype (*ab*) was more likely to fix than haplotype *aB*. The scale-free (*p* = 2) and star populations were always most likely to fix *ab* and least likely to fix *AB*.

The corresponding mean times to fixation are shown in figure 8. The shortest times across all simulations were for haplotype *ab*: if this initially common, least fit haplotype were to fix, it had to do so quickly, via genetic drift, before selection reduced its frequency too much. Such fixations took longer in regular populations than in scale-free (*p* = 2) and star structures. More recombination usually resulted in *AB* fixations taking longer, the exceptions being in panmictic populations subject to approximately equal selection strengths at each locus.

## 4. Discussion

More than 50 years ago Hill & Robertson [18] explored the dynamical behaviour of variation at two linked diallelic loci in a finite randomly mating diploid population and found that selection acting at one locus interfered with the ability of selection at the second locus to fix a favourable allele. This classic result became known as the Hill–Robertson effect and suggested that recombination had an evolutionary advantage in allowing faster adaptation [19]. Here we ask how the Hill–Robertson effect might apply to populations with spatial structure and explore the consequences of altering population structure from a regular ring and torus, through scale-free networks to a single centralized star. We compare all our results to those pertaining to a panmictic population.

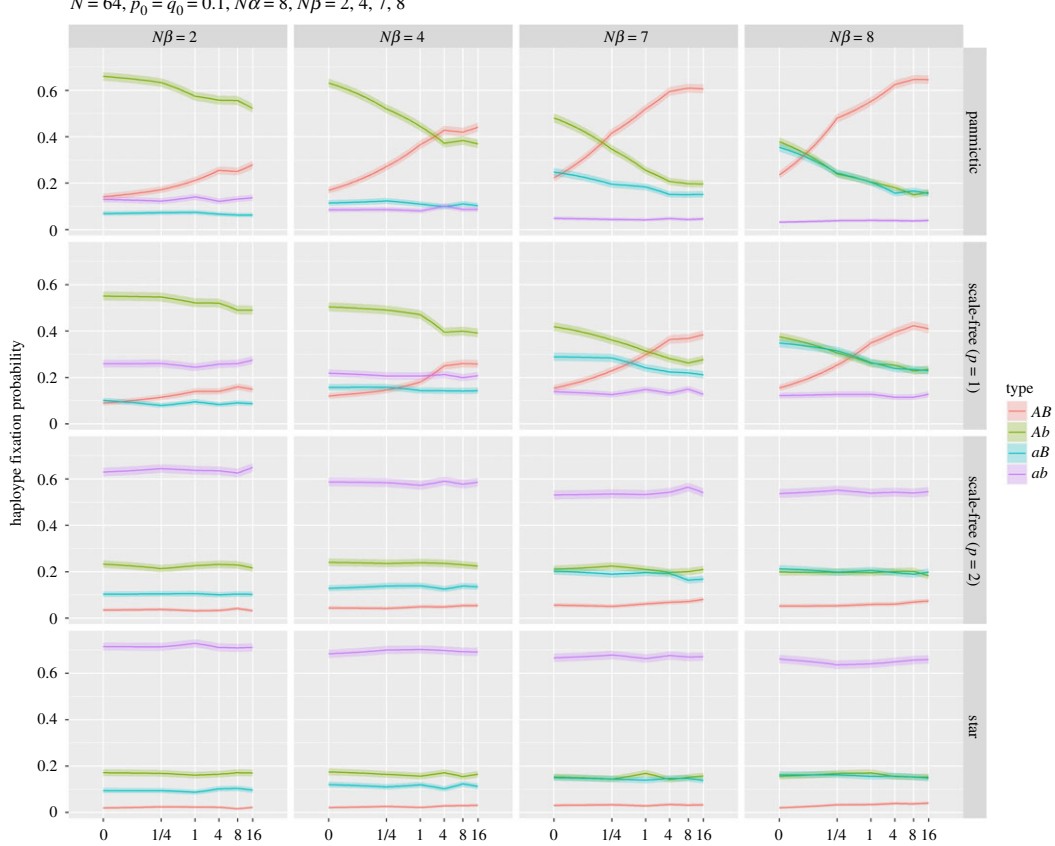

**Figure 7.** The probability of fixation of the four haplotypes for a range of linkage ($Nc$) when $p_0 = q_0 = 0.1$, $N = 64$ and $N\alpha = 8$. The ring and torus examples are not shown as they were indistinguishable from the panmictic network. The horizontal panels show selection at the second locus ($N\beta$) increasing from 2 to 8.

We find that:

1. The effect of population structure on the probability of fixation of a favourable allele is minimal for all regular networks, but for some scale-free (power 2) networks and the centralized star, this probability is greatly reduced. The main driver for this behaviour relates to graph degree variance.

2. By contrast, the mean time to fixation of the favoured allele is significantly affected: compared to a panmictic population, fixation takes much longer on a ring, torus and linear scale-free networks, and occurs much faster on scale-free networks ($p = 2$) and the star. Here fixation time is influenced by both graph path length and degree variance.

3. The likelihood with which each of the four haplotypes eventually fix is similar across panmictic, ring and torus populations, but scale-free populations are consistently less likely to fix the optimal haplotype (i.e. the one with the favoured allele at each locus). Moreover, in scale-free ($p = 2$) populations and the star, the time to fix any haplotype is much shorter than under other structures.

4. As Hill and Robertson found in panmictic populations, increasing recombination increases the likelihood of fixing the favoured $AB$ haplotype across all structures, although this effect was minimal with the scale-free ($p = 2$) and star. The time to fixation of $AB$, by contrast, usually increased with increasing recombination.

5. Star-like structures, with the largest graph degree variance, were overwhelmingly more likely to fix the least fit haplotype ($ab$) and did so significantly more rapidly than other population structures.

6. The invariance of fixation probability for regular structures (panmictic, ring and torus) suggest a form of isothermal theorem [10] may be true independent of the type of evolutionary model (generational versus steady-state) and form of genetic mixing and selection [38].

In addition, in contrast to Hill and Robertson's claim that if selection and recombination are scaled by population size ($N$) there is no effect of population size on the probability of fixation of the favourable allele, we find that very small panmictic populations are slightly less (or more) likely to fix this allele depending on selection strength, although the scaling appears to be true for larger $N$. Our study has benefited from the ability to run a far larger number of simulations than were possible over 50 years ago.

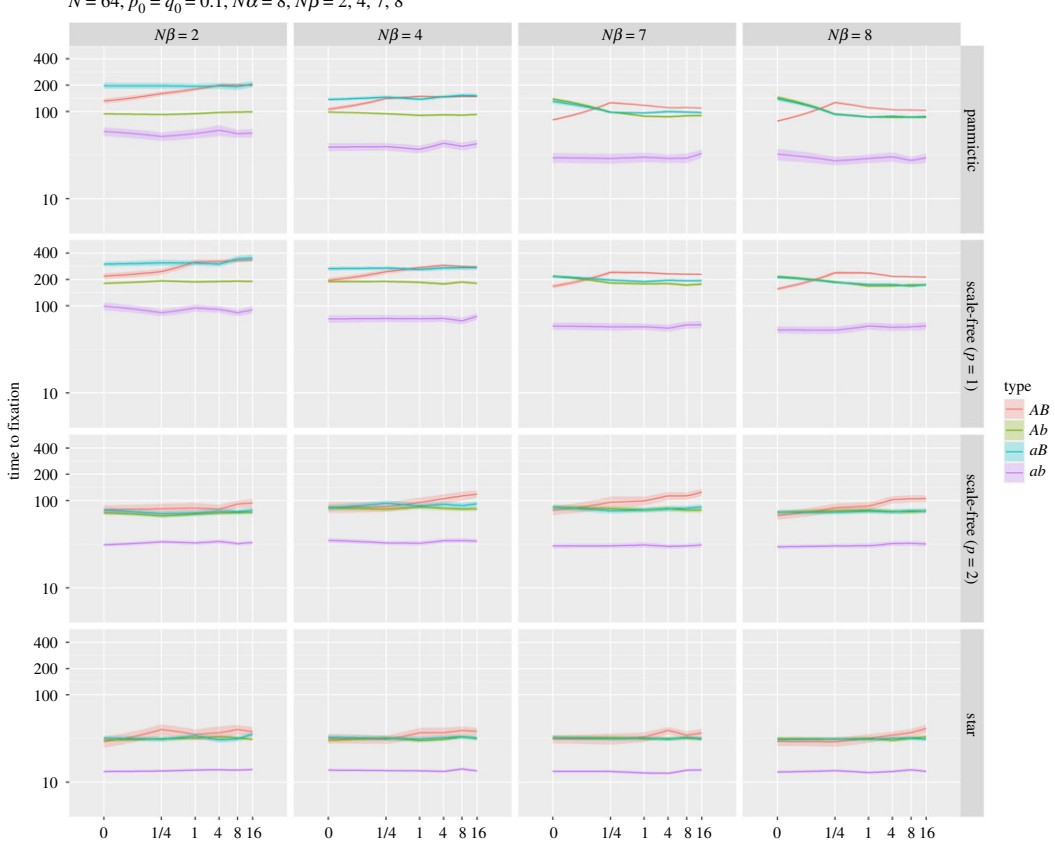

**Figure 8.** Mean time to fixation for the haplotypes for panmictic, scale-free ($p = 1$), scale-free ($p = 2$) and star networks. All parameters as for figure 7. Note $\log_{10}$ scale for y-axis. The complete set of network results are shown in the electronic supplementary material, figure S7.

The influence of recombination on the HR effect is clearly shown in figure 7. For example, when $N\alpha = 8$ and $N\beta = 4$, increasing the rate of recombination increases the fixation probability of the beneficial haplotype $AB$ for regular and scale-free ($p = 1$) populations. Of more interest is the behaviour when star-like spatial structures are introduced. As can be seen in figure 7, the influence of selection on the beneficial haplotype is significantly reduced, with a consequent increased chance of fixation of the initially most frequent but disadvantageous haplotype $ab$. In addition, figure 7 shows that the magnitude of the HR effect is reduced as the network becomes more star-like, since the difference in fixation probabilities reduces, irrespective of crossover rate.

Previous work examining network structures and fixation probability for the Moran process [10,12–15] showed that scale-free and star structures were apparently 'amplifiers of selection' (i.e. fixation of the favoured variant was more likely on such structures). Our results stand in stark contrast to these conclusions, with figures 3, 5 and 7 showing a clear reduction in fixation probability for the $A$ allele or the haplotype $AB$ for star-like structures compared with a panmictic population. In particular, the star graph shows $AB$ to be the *least* likely haplotype to fix for a wide range of parameter values. These different conclusions about the effect of population structure and the Moran process thus require some explanation. The Moran process is a model with overlapping generations with a single individual replaced in each time step (a so-called 'steady-state' evolutionary process [32]). A simple exploration of the role of generational versus steady-state models, and local versus global selection of parents, is presented in the electronic supplementary material. These models show that specific population dynamics are required to produce selection amplification for a star: a steady-state population with global selection and no parent replacement.

The panel $N\beta = 8$ (strong selection for $AB$) and large $Nc$ in figure 7 shows that the $AB$ haplotype is the most likely to fix for regular and linear scale-free graphs. However, under these conditions for the scale-free ($p = 2$) and star networks fixation of the $ab$ haplotype is most likely. Although the time to fixation for each haplotype has the same order for each network ($AB > Ab/aB > ab$), the star-like networks have a

greatly reduced fixation time. This suggests that star-like structures amplify drift, allowing the *ab* haplotype to fix prior to selection increasing the selected *AB* haplotype. Recent work [17] has examined the trade-off between fixation probability and time for the Moran process, to determine which population networks amplify selection while reducing fixation time. The results examined several types of bipartite graphs (graphs with two independent sets of vertices, U and V that are connected to each other, but no edges connect nodes within U or V) that exhibited these properties, which suggests future work to explicitly examine the HR effect on these population structures. However, since these results apply to the Moran process these properties may not, in light of our experiments, transfer across to a generational framework. This lack of generalization matters because of the suggested use of star structures for amplifying selection using directed evolution for enzymes and *in vitro* evolution [14,17], especially given that *in vitro* evolution methods are generational. In fact, it may well be that for a generational model regular population structure is the greatest amplifier of selection. A mathematical analysis of this behaviour is, however, beyond the scope of this paper.

The reduced fixation time due to star-like structures could be related to a reduction in effective population size ($N_e$) [39], although previous work [11] has shown that this parameter alone does not account for the relationship between ploidy and fixation under drift. Similarly, as shown with the normalized (panel *b*) model results of figures 3–6, for irregular graphs the behaviour of HR cannot be accounted for by adjustments to $N_e$ [24]. Rather, the structures themselves have an effect that interacts with the interplay between drift, recombination and selection.

Nevertheless, our findings are in broad agreement with Hill and Robertson. For example, across all our simulations, tighter linkage means that the favoured allele *A* at A is less likely to fix, especially when selection for *B* at B is stronger. Similarly, our results are also in accord with [11], which showed that fixation under drift at a single locus was faster as one moved from a ring to a scale-free structure.

To summarize, in the race to fixation between selection favouring fitter haplotypes and drift favouring common haplotypes, population structure and recombination interact to affect both the chances and speed of fixation.

Data accessibility. Data and relevant code for this research work are stored in GitHub: https://github.com/pwhigham/HR-model.git and have been archived within the Zenodo repository: https://doi.org/10.5281/zenodo.4362633.

Authors' contributions. H.G.S. conceived the proposal for the original research. Both authors designed the methods and contributed to the interpretation of the results and manuscript preparation. P.A.W. wrote the software for the models and figure construction. Both authors gave final approval for publication.

Competing interests. We declare we have no competing interest.

Funding. This work was not supported by a specific grant.

Acknowledgements. The authors would like to thank the reviewers for valuable comments, especially those that led to an improved understanding of the consequences of effective population size.

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
