## [Peer Review File · Royal Society Open Science]

Review History

RSOS-201831.R0 (Original submission)

Review form: Reviewer 1

Is the manuscript scientifically sound in its present form?

Yes

Are the interpretations and conclusions justified by the results?

Yes

Is the language acceptable?

Yes

Do you have any ethical concerns with this paper?

No

Have you any concerns about statistical analyses in this paper?

No

Recommendation?

Major revision is needed (please make suggestions in comments)

Comments to the Author(s)

This paper is not much changed from the original version I saw.

Having just read it carefully again, my opinion remains much as the first time: this is a fine paper that has believable results, but the population structures used (although previous used in another prominent paper) are not reflective of natural settings.

It seems at least necessary to comment about the unrealism of these chosen structures (or to defend them biologically beyond simply stating this is what another theory paper used).

More importantly though, given the lack of realism I would like the paper to try to unpack the results a bit more. We already know that what the paper calls "regular" structures (and what Nagylaki called conservative migration) do not affect N_e . And we know that star structures etc. do decrease N_e substantially. We also know that a lower N_e/N ratio will substantially affect the probability of fixation. The paper asserts that the effect observed are not just the results of the change in N_e , but it does not really defend this. If indeed this is true (and I'm happy to believe that it is), the extra effects seem trivial compared to the effect of N_e . But this comparison is never made. I would love to see a version of Figure 3 that demonstrated a scaled response to linked selection, expressed as a ratio of the case where the other locus has no selection (i.e. standardized by the probability of fixation with $N*\beta = 0$).

(This is what I was suggesting, perhaps unclearly, in my previous review.) The authors state that they show the Kimura results, which is good, but the claim of this paper is that linked selection affects the probability of fixation over and above the effects of the population structure on a single locus. OK, we already know that the population structure affects probability of fixation. We already know that linked selection affects probability of fixation. The new question is whether these two interact in any way, and the paper never actually shows that they do concretely. The paper actually alludes to the fact that the pattern is actually similar in reduction in proportional fixation probability across different structures (lines 19-21 on p9) but is doesn't show us concretely how much this is true. The possible large effect of N_e is referred to in the discussion (p16, lines 38 onwards), but it is odd that it is not actually calculated and investigated. I believe the core insight to be (possibly) gained from this work is potentially left almost unstated and unproven.

I would also like to see an acknowledgment that the literature already predicts that time to fixation should be longer with isolation by distance models.

Review form: Reviewer 2**Is the manuscript scientifically sound in its present form?**

Yes

Are the interpretations and conclusions justified by the results?

Yes

Is the language acceptable?

Yes

Do you have any ethical concerns with this paper?

No

Have you any concerns about statistical analyses in this paper?

No

Recommendation?

Accept with minor revision (please list in comments)

Comments to the Author(s)

This is an interesting and useful paper with little to criticise. The authors explore a range of population structures, recombination rates, population sizes and selection coefficients, revealing what I see as modest but potentially important deviations from the standard Hill-Robertson effect theory. Consequently, I have relatively few comments.

To me, the biggest shortfall is the lack of a targeted attempt to uncover the under-pinning mechanism. Maybe I am wrong, but surely the differences between the various population structures can be distilled into their component elements, population size and population connectivity? Larger populations are more likely to include rare products of recombination that mean both beneficial alleles lie on the same chromosome, after which selection will tend to do the rest. Equally, connectivity between populations will tend to influence the extent to which stochastic early events (a given population sub-unit carrying mainly chromosomes with the beneficial allele at one locus but not the other) allows the build-up of opposing outcomes in different sub-units. Higher connectivity (e.g. the star, where every sub-unit is within two migration events of any other) presumably allows a more homogeneous / deterministic process. I wonder whether this could be explored more systematically using population pairs and varying the migration rate between the two. Or is this more what previous work has done (I admit, this is not really my area). Nonetheless, it would be nice to see more progress into *why* the different population structures give such different results. Even changing the probability of mixing along an edge should be informative - what happens when a star is treated in this way?

Thus, while it is interesting to use different arrangements of populations, it seems that these are chosen more because they are easy to model than because they illuminate any particular aspect of the problem (or indeed mimic reality particularly well). An alternative approach to the systematic varying of parameters that we can be sure will impact key parameters like size and connectivity would be to measure these quantities. For example, repeat runs with neutral markers would allow an estimation of N_e and of F_{st} for each scenario. At minimum, this would allow the authors to be able to state whether or not these two variables can explain the variation in outcome or, alternatively, whether there is some further emergent property. If the former, job done! If the latter, this is equally interesting. I realise I am likely being over-picky, but as it stands I can't help feeling that the different population structures mask as much as they illuminate the key underlying processes.

Decision letter (RSOS-201831.R0)

Dear Professor Whigham

The Editors assigned to your paper RSOS-201831 "Graph-Structured Populations and the Hill-Robertson Effect" have now received comments from reviewers and would like you to revise the paper in accordance with the reviewer comments and any comments from the Editors. Please note this decision does not guarantee eventual acceptance.

Both reviewers raise substantive points that will need very careful consideration. We invite you to respond to the comments supplied below and revise your manuscript. Below the referees' and Editors' comments (where applicable) we provide additional requirements. Final acceptance of your manuscript is dependent on these requirements being met. We provide guidance below to help you prepare your revision.

Please submit your revised manuscript and required files (see below) no later than 21 days from today's (ie 02-Dec-2020) date. Note: the ScholarOne system will 'lock' if submission of the revision is attempted 21 or more days after the deadline. If you do not think you will be able to meet this deadline please contact the editorial office immediately.

on behalf of Professor Steve Brown (Associate Editor) and Steve Brown (Subject Editor)
openscience@royalsociety.org

Reviewer comments to Author:
Reviewer: 1

Comments to the Author(s)
This paper is not much changed from the original version I saw.

Having just read it carefully again, my opinion remains much as the first time: this is a fine paper that has believable results, but the population structures used (although previous used in another prominent paper) are not reflective of natural settings.

It seems at least necessary to comment about the unrealism of these chosen structures (or to defend them biologically beyond simply stating this is what another theory paper used).

More importantly though, given the lack of realism I would like the paper to try to unpack the results a bit more. We already know that what the paper calls “regular” structures (and what Nagylaki called conservative migration) do not affect N_e . And we know that star structures etc. do decrease N_e substantially. We also know that a lower N_e/N ratio will substantially affect the probability of fixation. The paper asserts that the effect observed are not just the results of the change in N_e , but it does not really defend this. If indeed this is true (and I’m happy to believe that it is), the extra effects seem trivial compared to the effect of N_e . But this comparison is never made. I would love to see a version of Figure 3 that demonstrated a scaled response to linked selection, expressed as a ratio of the case where the other locus has no selection (i.e. standardized by the probability of fixation with $N \cdot \beta = 0$).

(This is what I was suggesting, perhaps unclearly, in my previous review.) The authors state that they show the Kimura results, which is good, but the claim of this paper is that linked selection affects the probability of fixation over and above the effects of the population structure on a single locus. OK, we already know that the population structure affects probability of fixation. We already know that linked selection affects probability of fixation. The new question is whether these two interact in any way, and the paper never actually shows that they do concretely. The paper actually alludes to the fact that the pattern is actually similar in reduction in proportional fixation probability across different structures (lines 19-21 on p9) but it doesn’t show us concretely how much this is true. The possible large effect of N_e is referred to in the discussion (p16, lines 38 onwards), but it is odd that it is not actually calculated and investigated. I believe the core insight to be (possibly) gained from this work is potentially left almost unstated and unproven.

I would also like to see an acknowledgment that the literature already predicts that time to fixation should be longer with isolation by distance models.

Reviewer: 2

Comments to the Author(s)

This is an interesting and useful paper with little to criticise. The authors explore a range of population structures, recombination rates, population sizes and selection coefficients, revealing what I see as modest but potentially important deviations from the standard Hill-Robertson effect theory. Consequently, I have relatively few comments.

To me, the biggest shortfall is the lack of a targeted attempt to uncover the under-pinning mechanism. Maybe I am wrong, but surely the differences between the various population structures can be distilled into their component elements, population size and population connectivity? Larger populations are more likely to include rare products of recombination that mean both beneficial alleles lie on the same chromosome, after which selection will tend to do the rest. Equally, connectivity between populations will tend to influence the extent to which stochastic early events (a given population sub-unit carrying mainly chromosomes with the beneficial allele at one locus but not the other) allows the build-up of opposing outcomes in different sub-units. Higher connectivity (e.g. the star, where every sub-unit is within two migration events of any other) presumably allows a more homogeneous / deterministic process. I wonder whether this could be explored more systematically using population pairs and varying the migration rate between the two. Or is this more what previous work has done (I admit, this is not really my area). Nonetheless, it would be nice to see more progress into *why* the different population structures give such different results. Even changing the probability of mixing along an edge should be informative – what happens when a star is treated in this way?

Thus, while it is interesting to use different arrangements of populations, it seems that these are chosen more because they are easy to model than because they illuminate any particular aspect of the problem (or indeed mimic reality particularly well). An alternative approach to the systematic varying of parameters that we can be sure will impact key parameters like size and connectivity would be to measure these quantities. For example, repeat runs with neutral markers would allow an estimation of N_e and of F_{st} for each scenario. At minimum, this would allow the authors to be able to state whether or not these two variables can explain the variation in outcome or, alternatively, whether there is some further emergent property. If the former, job done! If the latter, this is equally interesting. I realise I am likely being over-picky, but as it stands I can't help feeling that the different population structures mask as much as they illuminate the key underlying processes.

===PREPARING YOUR MANUSCRIPT===

===PREPARING YOUR REVISION IN SCHOLARONE===

Author's Response to Decision Letter for (RSOS-201831.R0)

See Appendix A.

RSOS-201831.R1 (Revision)

Review form: Reviewer 1

Is the manuscript scientifically sound in its present form?

Yes

Are the interpretations and conclusions justified by the results?

Yes

Is the language acceptable?

Yes

Do you have any ethical concerns with this paper?

No

Have you any concerns about statistical analyses in this paper?

No

Recommendation?

Accept as is

Comments to the Author(s)

.

Review form: Reviewer 2

Is the manuscript scientifically sound in its present form?

Yes

Are the interpretations and conclusions justified by the results?

Yes

Is the language acceptable?

Yes

Do you have any ethical concerns with this paper?

No

Have you any concerns about statistical analyses in this paper?

No

Recommendation?

Accept with minor revision (please list in comments)

Comments to the Author(s)

I admit I am torn. I don't think I asked too much in my first review and I feel that the responding actions have been rather minimal. Consequently, my concerns remain that this is, to a large extent, a merely descriptive exercise that finds some interesting patterns in unrealistic population

structures and provides the basis for future research. As such, this work contributes very little to our understanding beyond showing that population structure is important, which we already know (the impact on N_e , phenomena like allele surfing etc. etc.). I have rather grudgingly ticked accept minor, as much because I can see they effort has been made to address the other Referee's comments ad because my suggestions have been addressed. However, I continue to believe that it is not am massive ask to obtain some measures of key characters such as connectivity and at least try to produce a general framework for understanding why different population structures behave as they do.

Decision letter (RSOS-201831.R1)

Dear Professor Whigham

On behalf of the Editors, we are pleased to inform you that your Manuscript RSOS-201831.R1 "Graph-Structured Populations and the Hill-Robertson Effect" has been accepted for publication in Royal Society Open Science subject to minor revision in accordance with the referees' reports. Please find the referees' comments along with any feedback from the Editors below my signature.

Both reviewers conclude that the manuscript should be accepted following your revisions. One reviewer makes a number of comments reflecting their particular enthusiasms. However, it will be up to you to consider and conclude what, if any, additional minor revisions you might wish to make.

Below the referees' and Editors' comments (where applicable) we provide additional requirements. Final acceptance of your manuscript is dependent on these requirements being met. We provide guidance below to help you prepare your revision.

Please submit your revised manuscript and required files (see below) no later than 7 days from today's (ie 03-Feb-2021) date. Note: the ScholarOne system will 'lock' if submission of the revision is attempted 7 or more days after the deadline. If you do not think you will be able to meet this deadline please contact the editorial office immediately.

on behalf of Professor Steve Brown (Subject Editor)
 openscience@royalsociety.org

Reviewer comments to Author:

Reviewer: 1

Comments to the Author(s)

.

Reviewer: 2

Comments to the Author(s)

I admit I am torn. I don't think I asked too much in my first review and I feel that the responding actions have been rather minimal. Consequently, my concerns remain that this is, to a large extent, a merely descriptive exercise that finds some interesting patterns in unrealistic population structures and provides the basis for future research. As such, this work contributes very little to our understanding beyond showing that population structure is important, which we already know (the impact on N_e , phenomena like allele surfing etc. etc.). I have rather grudgingly ticked accept minor, as much because I can see they effort has been made to address the other Referee's comments ad because my suggestions have been addressed. However, I continue to believe that it is not am massive ask to obtain some measures of key characters such as connectivity and at least try to produce a general framework for understanding why different population structures behave as they do.

===PREPARING YOUR MANUSCRIPT===

===PREPARING YOUR REVISION IN SCHOLARONE===

Author's Response to Decision Letter for (RSOS-201831.R1)

See Appendix B.

Decision letter (RSOS-201831.R2)

Dear Professor Whigham,

It is a pleasure to accept your manuscript entitled "Graph-Structured Populations and the Hill-Robertson Effect" in its current form for publication in Royal Society Open Science.

You can expect to receive a proof of your article in the near future. Please contact the editorial office (openscience@royalsociety.org) and the production office (openscience_proofs@royalsociety.org) to let us know if you are likely to be away from e-mail contact – if you are going to be away, please nominate a co-author (if available) to manage the proofing process, and ensure they are copied into your email to the journal.

on behalf of Professor Steve Brown (Subject Editor)
openscience@royalsociety.org

Follow Royal Society Publishing on Twitter: @RSocPublishing
Follow Royal Society Publishing on Facebook:
<https://www.facebook.com/RoyalSocietyPublishing.FanPage/>

Read Royal Society Publishing's blog:
<https://royalsociety.org/blog/blogsearchpage/?category=Publishing>

Appendix A

Dear Reviewers,

Thank you for the useful and thoughtful review comments which have improved the paper significantly. Please find below the changes that we have made regarding the issues you have raised.

Reviewer: 1

Comments to the Author(s)

Having just read it carefully again, my opinion remains much as the first time: this is a fine paper that has believable results, but the population structures used (although previous used in another prominent paper) are not reflective of natural settings. It seems at least necessary to comment about the unrealism of these chosen structures (or to defend them biologically beyond simply stating this is what another theory paper used).

It is true that the presented graphs do not represent any real populations. It is worth noting that a panmictic population is also unrealistic due to the spatial constraints of geography. However, a network (graph) can represent any possible configuration of deme structure and breeding constraints for a fixed configuration of a population. Hence although the presented graphs do not represent any known population structures (a problem that biologists struggle to address) we have used several extreme cases to exemplify the effects of irregular or highly constrained population structures. For example, the ring is one extreme example of a regular structure, while a star is an extreme of a single individual influencing an entire population. The two scale-free models show the effect of linear structures with increasing levels of clustering. This approach allows us to highlight how changes in population structure interact with the HR model. We have included the following statements in the Methods section to acknowledge the unrealistic nature of the graph structures used in the paper and our justification for the approach:

“To biologists the graph structures used in this paper may seem arbitrary and unrealistic. However, the structures are chosen to highlight and exemplify different population constraints. For example, the ring and star are extreme examples of a regular structure and single clustered population, while the scale-free graphs represent linear structures with increased local clustering. The purpose here is to emphasize how changing population structures affect and interact with linked loci. We acknowledge that biological populations will only present elements of these differences due to the complexity of a real population.”

More importantly though, given the lack of realism I would like the paper to try to unpack the results a bit more. We already know that what the paper calls “regular” structures (and what Nagylaki called conservative migration) do not affect N_e .

This is true for fixation probability but not for time to fixation, where N_e can be significantly affected (e.g. slower for rings).

And we know that star structures etc. do decrease N_e substantially. We also know that a lower N_e/N ratio will substantially affect the probability of fixation. The paper asserts that the effect observed are not just the results of the change in N_e , but it does not really defend this. If indeed this is true (and I’m happy to believe that it is), the extra effects seem trivial compared to the effect of N_e . But this comparison is never made. I would love to see a version of Figure 3 that demonstrated a scaled

response to linked selection, expressed as a ratio of the case where the other locus has no selection (i.e. standardized by the probability of fixation with $N\beta = 0$).
 (This is what I was suggesting, perhaps unclearly, in my previous review.)

The authors state that they show the Kimura results, which is good, but the claim of this paper is that linked selection affects the probability of fixation over and above the effects of the population structure on a single locus. OK, we already know that the population structure affects probability of fixation. We already know that linked selection affects probability of fixation. The new question is whether these two interact in any way, and the paper never actually shows that they do concretely. The paper actually alludes to the fact that the pattern is actually similar in reduction in proportional fixation probability across different structures (lines 19-21 on p9) but it doesn't show us concretely how much this is true. The possible large effect of N_e is referred to in the discussion (p16, lines 38 onwards), but it is odd that it is not actually calculated and investigated. I believe the core insight to be (possibly) gained from this work is potentially left almost unstated and unproven.

Thanks for the suggestion (which we had not understood previously) – this clearly helps to unpack the behaviour of the model. We have created additional normalised versions for Fig 3 – 6, which have been normalised either with $N\beta=0$ (for each q_0) or $N_c=0$ (for each $N\beta$). This has allowed a clear pattern of the contribution of selection, crossover and spatial structure to be decomposed. For example, Fig. 3 now shows the original and normalised versions of the results for the fixation probability of the A allele:

Panel B shows the model normalized by $N\beta=0$ for each q_0 and shows that until the population structures have nodes with extreme differences in degree (Scale-free $p=2$ and Star) that the HR behaviour is more-or-less the same. For the extreme structures the normalized model indicates that the influence of selection at the second locus is reduced, which is related to the time to fixation, as shown below (new version of Fig. 4):

A $N = 64, p_0 = 0.3, N\alpha = 4, N\beta = 0$

B $N = 64, p_0 = 0.3, N\alpha = 4, N\beta = 0$

Panel A shows the increased/reduced time to fixation due to increased/decreased path lengths. Panel B (normalized by the time for $N\beta = 0$ for each q_0) shows that when the selection strength on both alleles is equal ($N\alpha = N\beta = 4$) fixation time increases. For the star there is reduced difference between selection strengths because, if fixation is to occur, it must happen early in the process. The following sentences have been included for Figs 3 and 4:

“We also show, in Panel B of Figure 3, the results scaled by the response when there is no selection at the second locus. Here each graph is normalized by the probability of fixation when $N\beta=0$ for each q_0 . Panel B shows that relative HR probabilities are more-or-less the same for graphs that are regular or have some clustering with linear structures (Scale-Free $p=1$). When a small number of nodes have high degrees compared to most other nodes (Scale-Free $p=2$ and Star), however, the effect of linked selection is reduced, which cannot be accounted for by some sort of effective population-size adjustment. In other words, the effects we see are not simply the result of some change in effective population size: structure matters, too.”

“Figure 4 (panel B) shows normalized fixation time which shows two properties: as the path length of the graph increases (e.g., as it does in the ring), the relative fixation time increases for balanced selection ($N\alpha = N\beta$); for the scale-free ($p=2$) and star graphs the effect on fixation time due to different selection strengths is reduced. Again, these results show that a simple adjustment to effective population size would not account for the differences in fixation time.”

Figures 5 and 6 have been modified to show normalization due to crossover as:

A $N = 64, p_0 = q_0 = 0.1, N\alpha = 8$

B $N = 64, p_0 = q_0 = 0.1, N\alpha = 8$

A $N = 64, p_0 = q_0 = 0.1, N\alpha = 8$

B $N = 64, p_0 = q_0 = 0.1, N\alpha = 8$

The following additional statements have been made regarding figures 5 and 6:

“Panel B of Figures 5 and 6 show the normalized response to fixation and time to fixation by dividing the responses by the values for $Nc = 0$ for each $N\beta$. Both show that the extreme degree structures of the scale-free ($p=2$) and star result in responses that cannot be accounted for by effective population-size adjustments alone. Interestingly, for these graphs when $N\beta \leq N\alpha$ normalized fixation time is unaffected, irrespective of recombination rate.”

Thanks for these comments. We have endeavoured to use the normalised versions of fixation and time to argue that there is an interaction between the spatial structure and linked loci. These additional arguments are included above.

I would also like to see an acknowledgment that the literature already predicts that time to fixation should be longer with isolation by distance models.

The following sentence has been included:

“This result was first examined in detail by Slatkin (1981), where he showed that fixation time increased for both stepping stone and island-based models compared to the equivalent panmictic population.”

Reviewer: 2

Comments to the Author(s)

This is an interesting and useful paper with little to criticise. The authors explore a range of population structures, recombination rates, population sizes and selection coefficients, revealing what I see as modest but potentially important deviations from the standard Hill-Robertson effect theory. Consequently, I have relatively few comments.

To me, the biggest shortfall is the lack of a targeted attempt to uncover the under-pinning mechanism. Maybe I am wrong, but surely the differences between the various population structures can be distilled into their component elements, population size and population connectivity? Larger populations are more likely to include rare products of recombination that mean both beneficial alleles lie on the same chromosome, after which selection will tend to do the rest. Equally, connectivity between populations will tend to influence the extent to which stochastic early events (a given population sub-unit carrying mainly chromosomes with the beneficial allele at one locus but not the other) allows the build-up of opposing outcomes in different sub-units. Higher connectivity (e.g. the star, where every sub-unit is within two migration events of any other) presumably allows a more homogeneous / deterministic process. I wonder whether this could be explored more systematically using population pairs and varying the migration rate between the two. Or is this more what previous work has done (I admit, this is not really my area). Nonetheless, it would be nice to see more progress into **why** the different population structures give such different results. Even changing the probability of mixing along an edge should be informative – what happens when a star is treated in this way?

Thanks for these interesting ideas. The paper is the first real attempt to understand the HR effect for spatial structures (on graphs) which is why we specifically followed the original HR paper in terms of structure and model parameters. We have now addressed (to some extent) the underlying properties of the system through normalisation (see comments, Reviewer 1). Varying the underlying connectivity probabilities (effectively migration rates) for graphs such as a star are possible future research. We agree that there is further work here to pull apart the ways in which the spatial structure and linked loci interact, but this would be a significant examination of one space and varying its properties and those of a neutral/linked model. This seems more appropriate as future work, reflecting on the current results.

Thus, while it is interesting to use different arrangements of populations, it seems that these are chosen more because they are easy to model than because they illuminate any particular aspect of the problem (or indeed mimic reality particularly well).

We have included some additional justification for the use of the graphs in the paper (see Review 1 comments). The use of extreme spatial cases (such as the ring and star) seem justified to exemplify any behaviours that might occur for specific types of neighbourhood relationships.

An alternative approach to the systematic varying of parameters that we can be sure will impact key parameters like size and connectivity would be to measure these quantities. For example, repeat runs with neutral markers would allow an estimation of N_e and of F_{st} for each scenario. At minimum, this would allow the authors to be able to state whether or not these two variables can explain the variation in outcome or, alternatively, whether there is some further emergent property. If the former, job done! If the latter, this is equally interesting. I realise I am likely being over-picky, but as it stands I can't help feeling that the different population structures mask as much as they illuminate the key underlying processes.

We have addressed these issues (to some extent) with the normalization of results for fixation probability and time to fixation, which suggest some unusual behaviour for certain combinations of selection/crossover at both loci.

Appendix B

Dear Reviewers,

Thank you for the useful and thoughtful review comments which have helped to clarify the relationship between graph structure and fixation. Please find below the changes that we have made regarding the issues you have raised.

Reviewer: 1

No comments.

Reviewer: 2

However, I continue to believe that it is not an massive ask to obtain some measures of key characters such as connectivity and at least try to produce a general framework for understanding why different population structures behave as they do.

Thanks for the comments and yes, we agree that a general framework would be useful. We have addressed some additional issues with graph structure by examining the degree variance (i.e. the amount of irregularity in the graph) for each graph. Of course, for regular graphs this is zero, while for the scale free and star graphs the variance increases. This property is correlated with the changes in fixation probability. Although fixation time is dependent on path length for regular graphs it is also clear that degree variance contributes to fixation time. Comments regarding the influence of graph degree variance have been included in the results and discussion.

The following has been added to section 3.2:

The variance in degree for a graph can be used as a general measure of graph irregularity. For all regular graphs this value is zero, while for scale-free ($p=1$) degree variance is ~ 3.7 , scale-free ($p=2$) degree variance is ~ 15.6 and for a star degree variance is 60.06. This clearly correlates to the observed changes in fixation probability and implies that graphs with nodes that are much more connected than average are strong influences of fixation.

Some short additional comments in the discussion have been included highlighting the roles of graph path length and degree variance. See REVISED version.